# Frozen Elephant Trunk for Aortic Dissection Using Different Hybrid Grafts: Preliminary Results from a Prospective Study

**DOI:** 10.3390/jpm13050784

**Published:** 2023-04-30

**Authors:** Boris Kozlov, Dmitri Panfilov, Vitaliy Lukinov

**Affiliations:** 1Cardiology Research Institute—Branch of the Federal State Budgetary Scientific Institution ‘Tomsk National Research Medical Center of the Russian Academy of Sciences’, 634012 Tomsk, Russia; bnkozlov@yandex.ru; 2Institute of Computational Mathematics and Mathematical Geophysics, Siberian Branch of the Russian Academy of Sciences, 630090 Novosibirsk, Russia

**Keywords:** thoracic aorta, aortic dissection, hybrid graft, frozen elephant trunk, mortality

## Abstract

Background. The frozen elephant trunk technique has become popular and led to an expansion of indications for surgery. Various hybrid grafts for the frozen elephant trunk are used, sometimes with significantly different features. The objective of this study was to compare early- and mid-term outcomes after the frozen elephant trunk for aortic dissection using different hybrid grafts. Methods. The prospective study included 45 patients with acute/chronic aortic dissections. The patients were randomized into two groups. Group 1 patients (n = 19) were implanted with a hybrid graft E-vita open plus (E-vita OP). Group 2 (n = 26) included patients who received a MedEng graft. The inclusion criteria were type A and type B acute and chronic aortic dissection. The exclusion criteria were as follows: hyperacute aortic dissection (less than 24 h), organ malperfusion, oncology, severe heart failure, stroke, and acute myocardial infarction. The primary endpoint was early- and mid-term mortality. The secondary endpoints were postoperative complications (stroke and spinal cord ischemia, myocardial infarction, respiratory failure, acute renal injury, and re-operation for bleeding). Results. The rate of stroke and spinal cord ischemia in the E-vita OP vs. MedEng groups was 11% vs. 4% (*p* = 0.565) and 11% vs. 0% (*p* = 0.173), respectively. The respiratory failure rate was comparable in both groups (*p* > 0.999). Acute kidney injury requiring hemodialysis and the need for re-sternotomy in the MedEng group vs. E-vita OP group was 31% vs. 16% (*p* = 0.309) and 15% vs. none (*p* = 0.126), respectively. Early mortality in the MedEng and E-vita OP groups did not differ (8% vs. 0, *p* = 0.501). The mid-term survival in the analyzed groups was 79% vs. 61%, (*p* = 0.079), respectively. Conclusions. No statistically significant differences were observed between patients receiving frozen elephant trunk with the hybrid MedEng and E-vita OP grafts in regard to early mortality and morbidity. Mid-term survival was also non-significant between analyzed groups with a trend toward more favorable mortality in the MedEng group.

## 1. Introduction

Until recently, open surgery has been the priority form of care in thoracic aortic surgery [1]. For a long period of time, the conventional ‘elephant trunk’ (cET) technique proposed by Borst H.G. in 1983 has remained the ‘gold standard’ [2]. However, in the mid-1990s in Asia and in the early 2000s in Europe, the frozen elephant trunk (FET) technique, which combines conventional aortic arch surgery and the endovascular repair of the descending aorta, was proposed [3,4]. A literature assessment of the efficacy of cET and FET demonstrated satisfactory results for the new technique, especially in aortic dissection [5]. This condition occurs in 2.6–7.2 people per 100,000 patient-years. The prognosis of patients with aortic dissection is poor if left untreated [6].

Taking this into account, frozen elephant trunk techniques have become popular and led to an expansion of uses for the FET technique [7]. A comprehensive analysis of the early and late results of this technique revealed its high efficacy [8,9,10]. As a result, this technique is included in the aortic surgeon armamentarium, which is reflected in current guidelines for the diagnosis and management of aortic disease [11].

To date, various modifications of hybrid grafts for the FET have been used, sometimes with significantly different features. From a technical perspective, among commercially available FET hybrid grafts, each of them has specific shortcomings. For instance, the deployment of the Thoraflex and E-vita open plus grafts is time-consuming and technically demanding. Moreover, the stented portion of the Thoraflex graft is not available in sizes of less than 28 mm. The Cronus and J grafts require a separate proximal graft and are not available in sizes of more than 32 mm. Additionally, these devices, as well as E-vita open plus, have no perfusion branches [12]. Since 2019, it has become possible to use the first Russian hybrid prosthesis for hybrid aortic arch surgery. To date, no comparative study has been conducted to evaluate the efficacy of the FET using a domestic hybrid graft. The aim of this study was to compare early and mid-term outcomes of the frozen elephant trunk technique for aortic dissection using different hybrid grafts.

## 2. Materials and Methods

Starting in January 2020, the prospective study (registration number ACTRN 12620000123943) included 45 patients with acute/chronic aortic dissection who were eligible for surgical treatment. The inclusion criteria were type A and type B acute and chronic aortic dissection. The exclusion criteria were as follows: hyperacute aortic dissection (less than 24 h), organ malperfusion, oncology, severe heart failure, stroke, acute myocardial infarction. The primary endpoint was early- and mid-term mortality. The secondary endpoints were postoperative complications (stroke and spinal cord ischemia, myocardial infarction, respiratory failure, acute renal injury, re-operation for bleeding).

The frozen elephant trunk procedure, in all cases, was performed through median sternotomy under lower body circulatory arrest with moderate hypothermia (25–28 °C) and unilateral cerebral antegrade perfusion via innominate artery. Patients were allocated into two groups in the ‘randomizeR’ package via a block randomization procedure with forty blocks; the length of one block was set to 4, the group ratio inside 1 block was set to 1:1, and the initial seed was set to 10. Group 1 (E-vita OP, n = 19) included patients who were implanted with an E-vita open plus (Jotec, Germany) hybrid graft with a diameter of 24–30 mm and a length of 150 mm. Group 2 (MedEng, n = 26) included patients who underwent the FET procedure using a MedEng hybrid graft (MedEng, Penza, Russia) with a diameter of 26–30 mm and a length of 150 mm (Figure 1).

The study was approved by the local ethics committee. Written informed consent was obtained from all patients.

### 2.1. Image Analysis

All measurements were taken using electrocardiography-gated computed tomographic angiography. Analysis was performed using the 64-slice scanner Discovery NM-CT 570c (GE Healthcare, Milwaukee, WI, USA) with an angiographic phase spatial resolution ranging from 0.6 to 1.25 mm. The adopted computed tomographic protocol included unenhanced, arterial, and delayed data acquisition. The arterial phase was acquired after the intravenous injection of 80–100 mL of s nonionic iodinated contrast at 5 mL/s, followed by a 50 mL bolus of saline solution. Delayed-phase scans were obtained 120–180 s after contrast injection. All measurements were taken in multiplanar reconstruction, always in the plane perpendicular to the manually corrected local aortic center line. Analysis and assessment of the images were based on the consensus between two experienced investigators.

### 2.2. Surgical Technique

Details of the FET procedure have been described elsewhere [12].

Briefly, after establishing a cardiopulmonary bypass, body cooling was started. When the target temperature was achieved, lower body circulatory arrest and antegrade cerebral perfusion were initiated. Then, the aorta was transected. After visual assessment of the aortic lumen, the hybrid graft was implanted into the true lumen of the descending aorta. After graft deployment in the descending aorta, it was fixed to the aorta via continuous suture using the sandwich technique. We did not oversize the hybrid graft in either acute or chronic AD. Usually, we perform the distal aortic anastomosis in Z2 or Z3. It depends on anatomical features and the technical complexity of the surgery. After distal anastomosis completion, lower body perfusion was started. During the rewarming of the patient’s body up to 36 °C, we re-implanted the supra-aortic vessels using the island technique, the separate re-implantation of the aortic arch vessels, or a combined technique. Then, the cardiopulmonary bypass was reinstituted. The proximal aortic reconstruction was performed during rewarming, as well as concomitant cardiac procedures if necessary, including coronary artery bypass grafting, aortic valve replacement, or the Bentall procedure.

During FET surgery, we followed a standard protocol adopted at our center, which is aimed at preventing spinal cord ischemia (SCI). It includes unilateral antegrade cerebral perfusion via the innominate artery with a flow rate of 8–10 mL/kg/min, a perfusion pressure of 60–80 mmHg, moderate hypothermia (25–28 °C), hematocrit values above 25%, and the maintenance of a mean pressure of 80–100 mmHg for the first 3 days after the surgery. Our monitoring strategy included arterial pressure measurements in both radial and femoral arteries and cerebral neuromonitoring using near-infrared spectroscopy. We did not perform cerebrospinal fluid drainage and pressure monitoring during the frozen elephant trunk procedure.

Follow-up was performed according to the institutional database, supplemented by individual patient records. Clinical and radiologic follow-ups were performed for all discharged patients upon discharge, at 6 and 12 months from the surgery, and annually thereafter. No patients were lost to follow-up. The median follow-up period was 12 (6–12) months. 

### 2.3. Statistical Analysis

Statistical calculations were performed using Rstudio 1.0.136 (RStudio, Inc., Boston, MA, USA), version 3.3.1. Distributions of the variables were evaluated for normality using the Shapiro–Wilk test. Continuous data that follow a normal distribution are described as the mean ± standard deviation; the ones that do not follow a normal distribution are described as a sample median and the 25th and 75th percentiles. Categorical variables are summarized as n (%). Obtained data were compared using the Mann–Whitney U-test for continuous variables and chi-square tests for categorical variables (Fisher’s exact tests when necessary due to small cell sizes). A survival analysis was created using the Kaplan–Meier method. Statistical differences were assessed using the log-rank test. Using univariate logistic regression analysis, significant variables for in-hospital survival were initially identified (*p* < 0.05) and included in the final multivariate logistic regression model to determine the independent risk factors. Statistical significance was defined as *p* < 0.05.

## 3. Results

There were no significant differences between the groups with respect to the analyzed preoperative variables. Demographics, comorbidity were comparable between the groups. The details are presented in Table 1.

Intraoperative data, including the operation and ischemic times and the complexity of the concomitant cardiac surgery, did not reveal statistical differences in analyzed groups of patients. Additionally, the technique of reconstruction of the aortic arch vessels was similar in both groups. There were no statistically significant differences in the proximal landing zone of the graft in the analyzed groups. However, the MedEng graft was implanted more frequently in zone 2 (42% vs. 32%, *p* = 0.543), and the E-vita OP hybrid prosthesis was implanted in zone 3 (58% vs. 42%, *p* = 0.373) (Table 2).

### 3.1. Early Results 

The in-hospital outcomes and complications are summarized in Table 3.

There were no significant differences in stroke, delirium, and paraplegia in either group. However, it is worth noting that delirium prevailed in the MedEng group, while stroke was more common in the E-vita OP group. It should be noted that there was no relationship between the aortic arch reconstruction technique and the incidence of neurological deficits.

The incidence of SCI also had no statistically significant differences. At the same time, this complication was observed in the E-vita OP group only (2 (11%) versus 0, *p* = 0.173). It is important to note that the distal landing zone level was similar in both groups (Figure 2).

The duration of lung ventilation and frequency of respiratory failure requiring tracheostomy were comparable in both groups. The incidence of acute kidney injury requiring hemodialysis was more common in the MedEng group (31% vs. 16%) but there was no statistical difference between the groups (*p* = 0.309). Drainage output volume was non-significantly higher in the MedEng group. There was no statistically significant difference in the re-sternotomy rate and the need for blood transfusions in the MedEng group compared to the E-vita group.

There were no statistical differences in the 30-day mortality of the analyzed groups of patients (*p* = 0.501). It is worth noting that deaths occurred in the E-vita OP group only. The causes of early death were myocardial infarction (n = 1) and uncontrollable bleeding (n = 1). There were no aortic-related deaths in the entire cohort.

### 3.2. Mid-Term Results 

A total of seven deaths were recorded during the follow-up period. Among the patients, two deaths were in the MedEng group, and five deaths were in the E-vita OP group. The cause of death was multiple organ failure in all of the patients. The mid-term survival rates for patients in the MedEng group and the E-vita OP group were 79% vs. 61% (*p* = 0.075), respectively (Figure 3). There were no aortic-related deaths.

### 3.3. Logistic Regression Analysis for Early Mortality 

Among the number of preoperative and intraoperative variables, the univariate analysis revealed that respiratory failure (OR = 24.89, 95%CI: 4.64–181.98, *p* < 0.001), acute kidney injury (OR = 9, 95%CI: 1.94–48.32, *p* = 0.006), the duration of surgery (OR = 1.01, 95%CI: 1–1.02, *p* = 0.014), cardiopulmonary bypass time (OR = 1.01, 95%CI: 1–1.03, *p* = 0.025), cardiac arrest time (OR = 1.02, 95%CI: 1–1.03, *p* = 0.039), and bleeding (OR = 14.57, 95%CI: 1.61–320.26, *p* = 0.029) were the risk factors of early mortality. These variables were included in the final multivariate regression model, which did not show any independent risk factors for early mortality.

## 4. Discussion

The frozen elephant trunk procedure is one of the options in the surgical treatment of aortic dissection [13;14]. The possibility of a single-stage procedure is one of the main advantages of this technique [13]. Thoracic aortic surgery using the FET technique provides satisfactory early- and mid-term results [5,14,15]. According to the literature, in-hospital mortality is up to 15% [9,10,16,17], and the 5-year survival rate exceeds 80% [14]. At the same time, the need for distal aortic re-interventions, mainly endovascular, is around 10–15% [13,15,18,19]. The increased popularity of the FET technique has encouraged manufacturers to produce different types of hybrid devices.

The first frozen elephant trunk procedure in Europe was performed in 2005 using the E-vita OP hybrid graft. To date, different grafts have been used for FET surgery all over the world [12,20,21]. For instance, Ius et al. [22] published their analysis of FET surgery using the Chavan-Haverich, E-vita OP, and Thoraflex prostheses. The authors demonstrated a comparable rate of morbidity and in-hospital and mid-term mortality, as well as the need for distal aortic re-interventions. Similarly, Berger et al. [13] did not show statistically significant differences in morbidity and early mortality rates after FET procedures using the E-vita OP and Thoraflex grafts. Furthermore, the authors found a statistically significantly greater need for endovascular re-interventions in the Thoraflex group (*p* = 0.003). Moreover, Harky et al. [23] published meta-analysis results showing that mortality and morbidity are lower in the E-vita OP graft versus the Thoraflex graft.

For a long period of time, the only hybrid graft used in Russia was the E-vita OP. However, in 2019, the first domestic hybrid graft designed for FET surgery was introduced. The first clinical experience showed good results utilizing the domestic device [24]. The experienced usage of the last two aforementioned grafts allowed us to conduct a study aimed at comparing early- and mid-term outcomes of the FET technique for aortic dissection using different hybrid grafts.

Analyzing the results, no significant differences were found for a number of outcomes despite the various design and technical features of the grafts. At the same time, in our opinion, the MedEng hybrid graft has a higher number of positive characteristics compared with the E-vita OP. Firstly, the deployment of the hybrid graft into the descending aorta is more convenient because of a short and flexible delivery system. Furthermore, the MedEng graft has a simpler mechanism for opening the stent graft. Secondly, the presence of the collar facilitates performing the distal aortic anastomosis. One of the important advantages of the MedEng hybrid graft is the presence of a perfusion branch, which provides lower body perfusion after the completion of a distal aortic anastomosis. At the same time, the E-vita OP hybrid graft lacks a perfusion branch, which creates certain intraoperative difficulties in providing lower body perfusion. Distal aortic perfusion provides sufficient blood flow for protection not only of the intestines, kidneys, and lower extremities but also the spinal cord. The last statement may be supported by our results regarding the incidence of spinal cord injuries. This complication was diagnosed in two patients in the E-vita OP group and none in the MedEng group.

According to our experience, the E-vita OP and MedEng grafts have similar features for left subclavian artery revascularization, especially in cases of implanting the graft into zone 2. It should be noted that the proximalization of the distal aortic anastomosis to zone 2 does not worsen the exposure of the proximal segment of the left subclavian artery, which could be accompanied by technical difficulties. At the same time, the displacement of the distal anastomosis proximally can shorten the operative ischemic times and reduce the extent of the surgery, which, in turn, improves mortality and morbidity in the early postoperative period [19,25,26,27,28]. Our analysis of the efficacy of the frozen elephant trunk procedure using different hybrid grafts is not limited by an assessment of the postoperative mortality and morbidity rate. The additional goals of further research are to study the behavior of the thoracoabdominal aorta in late follow-up, including the remodeling of the true, false, and total aortic lumen and the incidence of distal stent-graft-induced entries after FET surgery using the E-vita and MedEng hybrid grafts. 

### Limitations and Strengths

This study has limitations, including a single-center experience and a small study population. Additionally, the relatively short duration of the follow-up may not represent a full picture of the follow-up. These circumstances do not allow for high-quality conclusions. At the same time, the strength of the study is the prospective design, and this study is the first step toward an in-depth analysis of the issue discussed in this paper.

## 5. Conclusions

No statistical differences were observed between patients receiving frozen elephant trunk with the hybrid MedEng and E-vita OP grafts in regard to early mortality and morbidity. Mid-term survival was also non-significant among the analyzed groups, with a trend to more favorable mortality in the MedEng group.

## Figures and Tables

**Figure 1 jpm-13-00784-f001:**
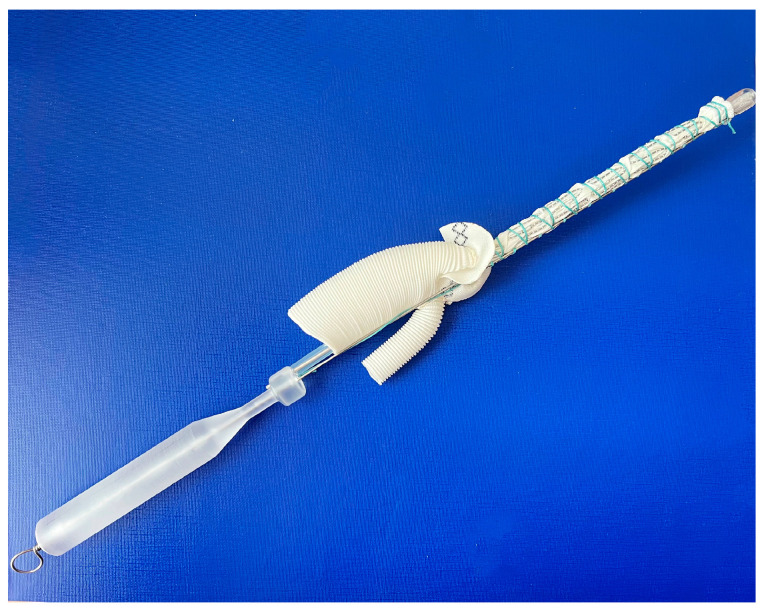
The MedEng hybrid stent graft.

**Figure 2 jpm-13-00784-f002:**
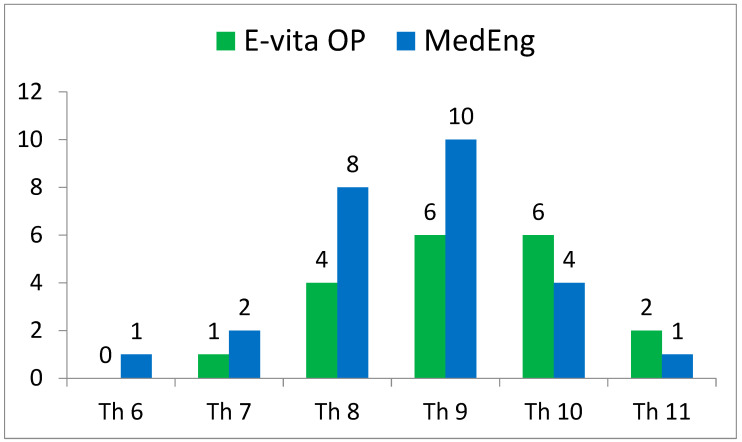
Distribution of the distal landing zone of a hybrid stent graft in the analyzed groups according to CT scans.

**Figure 3 jpm-13-00784-f003:**
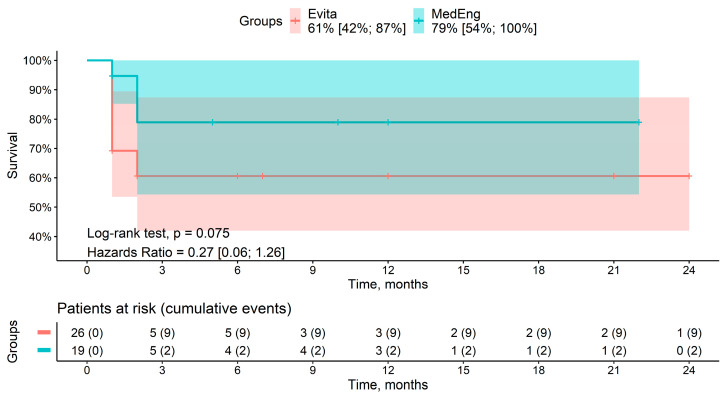
Mid-term survival rate of the patients in the analyzed groups.

**Table 1 jpm-13-00784-t001:** Preoperative data.

Variable	E-Vita OP (n = 19)	MedEng (n = 26)	Differences	*p*-Value
Height, cm	171.89 ± 10.13	169.04 ± 7.23	pseudo-MED (95%CI): −2 [−9; 3] SMD (95%CI): 0.33 [−0.28; 0.94]	0.333
Weight, kg	77.95 ± 13.1	82.04 ± 17.87	pseudo-MED (95%CI): 4 [−6; 12] SMD (95%CI): −0.26 [−0.86; 0.35]	0.557
BMI	26.77 ± 3.87	28.43 ± 5.25	pseudo-MED (95%CI): 1.29 [−2.19; 4.69] SMD (95%CI): −0.35 [−0.96; 0.25]	0.509
BSA	1.9 ± 0.18	1.9 ± 0.2	pseudo-MED (95%CI): 0.02 [−0.13; 0.14] SMD (95%CI): −0.02 [−0.67; 0.62]	0.906
Female sex, n (%)	5 26% [12%; 49%]	7 27% [14%; 46%]	RD: −1% [−27%; 26%] OR: 1 [0.2; 5]	>0.999
Age, n (%)	61.53 ± 9.23	58 ± 12.91	pseudo-MED (95%CI): −2 [−9; 4] SMD (95%CI): 0.31 [−0.29; 0.9]	0.534
Acute AD, n (%)	7 37% [19%; 59%]	13 50% [32%; 68%]	RD: −13% [−42%; 16%] OR: 1.7 [0.4; 6.9]	0.545
Chronic AD, n (%)	12 63% [41%; 81%]	12 46% [29%; 65%]	RD: 17% [−12%; 46%] OR: 0.5 [0.1; 2]	0.366
Type A AD, n (%)	9 47% [27%; 68%]	11 42% [26%; 61%]	RD: 5% [−24%; 34%] OR: 0.8 [0.2; 3.2]	0.770
Type B AD, n (%)	3 16% [6%; 38%]	7 27% [14%; 46%]	RD: −11% [−35%; 13%] OR: 1.9 [0.4; 13.5]	0.481
Type nonA-nonB AD, n (%)	1 5% [1%; 25%]	5 19% [9%; 38%]	RD: −14% [−32%; 4%] OR: 4.2 [0.4; 213.5]	0.222
Redo cardiac surgery, n (%)	1 5% [1%; 25%]	2 8% [2%; 24%]	RD: 3% [1%; 10%] OR: 0 [0; 28.5]	0.512
Hypertension, n (%)	16 84% [62%; 94%]	24 92% [76%; 98%]	RD: −8% [−27%; 11%] OR: 2.2 [0.2; 29.2]	0.636
Type 2 diabetes mellitus, n (%)	0 0% [0%; 17%]	2 8% [2%; 24%]	-	0.501
COPD, n (%)	1 5% [1%; 25%]	3 12% [4%; 30%]	RD: −7% [−23%; 9%] OR: 2.4 [0.2; 135.8]	0.622
Atrial fibrillation, n (%)	2 11% [3%; 31%]	3 12% [4%; 29%]	RD: −1% [−19%; 17%] OR: 1.1 [0.1; 14.6]	>0.999
History of stroke, n (%)	0	2 8% [2%; 24%]	-	0.501
CAD, n (%)	4 21% [9%; 43%]	8 31% [17%; 50%]	RD: −10% [−35%; 16%] OR: 1.6 [0.4; 9]	0.517
Creatinine, n (%)	104.06 ± 28.63	122.65 ± 103.44	pseudo-MED (95%CI): −4 [−19; 20] SMD (95%CI): −0.23 [−0.87; 0.41]	0.753
LVEF, %	65 [59.5; 65.5]	64 [61; 66]	pseudo-MED (95%CI): 0 [−3; 3] SMD (95%CI): 0.04 [−0.56; 0.63]	0.943
Severe aortic insufficiency, n (%)	1 5% [1%; 25%]	5 19% [9%; 38%]	RD: −14% [−32%; 4%] OR: 4.2 [0.4; 213.5]	0.222

AD—aortic dissection, BMI—body mass index, BSA—body surface area, CAD—coronary artery disease, COPD—chronic obstructive pulmonary disease, LVEF—left ventricular ejection fraction, pseudo-MED—pseudo-median-based procedure for comparing two independent groups, RD—risk difference, SMD—standardized mean difference.

**Table 2 jpm-13-00784-t002:** Intraoperative data.

Variable	E-Vita OP (n = 19)	MedEng (n = 26)	Differences	*p*-Value
Duration of the surgery, min	360 [300; 420]	390 [305; 505]	pseudo-MED (95%CI): 40 [−20; 120] SMD (95%CI): −0.5 [−1.17; 0.16]	0.196
CPB time, min	180 [165; 245]	205 [175; 270]	pseudo-MED (95%CI): 25 [−5; 65] SMD (95%CI): −0.53 [−1.16; 0.09]	0.112
Cardiac arrest time, min	144.5 [116.25; 198.75]	150 [137; 220]	pseudo-MED (95%CI): 13 [−18; 44] SMD (95%CI): −0.22 [−0.83; 0.39]	0.375
Lower body circulatory arrest time, min	28 [12; 34]	30 [20; 35]	pseudo-MED (95%CI): 1 [−5; 8] SMD (95%CI): −0.14 [−0.74; 0.45]	0.635
Cerebral perfusion time, min	67.5 [54.25; 78.75]	63 [48.75; 78.5]	pseudo-MED (95%CI): −5 [−20; 9] SMD (95%CI): 0.21 [−0.4; 0.82]	0.516
*Concomitant surgery*				
CABG, n (%)	3 16% [6%; 38%]	5 19% [9%; 38%]	RD: −3% [−26%; 19%] OR: 1.3 [0.2; 9.4]	>0.999
AVR, n (%)	0	1 4% [1%; 19%]	-	>0.999
Bentall procedure, n (%)	4 21% [9%; 43%]	3 12% [4%; 29%]	RD: 10% [−13%; 32%] OR: 0.5 [0.1; 3.4]	0.433
*Proximal landing zone*				
Zone 0, n (%)	1 5% [1%; 25%]	3 12% [4%; 29%]	RD: −6% [−22%; 10%] OR: 2.3 [0.2; 129.9]	0.627
Zone 1, n (%)	1 5% [1%; 25%]	1 4% [1%; 19%]	RD: 1% [−11%; 14%] OR: 0.7 [0; 59.6]	>0.999
Zone 2, n (%)	6 32% [15%; 54%]	11 42% [26%; 61%]	RD: −11% [−39%; 18%] OR: 1.6 [0.4; 6.8]	0.543
Zone 3, n (%)	11 58% [36%; 77%]	11 42% [26%; 61%]	RD: 16% [−14%; 45%] OR: 0.5 [0.1; 2.1]	0.373
*Supra-aortic vessel reconstruction*				
Island technique, n (%)	17 89% [69%; 97%]	18 69% [50%; 83%]	RD: 20% [−2%; 43%] OR: 0.3 [0; 1.6]	0.154
Island technique with separate re-implantation of the LSA, n (%)	6 32% [15%; 54%]	8 31% [17%; 50%]	RD: 1% [−27%; 28%] OR: 1 [0.2; 4.3]	>0.999
Separate re-implantation of the supra-aortic vessels, n (%)	2 11% [3%; 31%]	7 27% [14%; 46%]	RD: −16% [−38%; 6%] OR: 3.1 [0.5; 34.1]	0.264

AVR—aortic valve replacement, CABG—coronary artery bypass grafting, CPB—cardiopulmonary bypass, pseudo-MED—pseudo-median-based procedure for comparing two independent groups, LSA—left subclavian artery, RD—risk difference, SMD—standardized mean difference.

**Table 3 jpm-13-00784-t003:** Postoperative data.

Variable	E-Vita OP (n = 19)	MedEng (n = 26)	Differences	*p*-Value
ICU stay, days (range)	17.59 ± 25.06 (1–83)	8.33 ± 7.3 (1–34)	pseudo-MED (95%CI): 0 [−7; 4] SMD (95%CI): 0.53 [−0.12; 1.18]	0.965
Delirium, n (%)	0	4 15% [6%; 34%]	-	0.126
Stroke, n (%)	2 11% [3%; 31%]	1 4% [1%; 19%]	RD: 7% [−9%; 22%] OR: 0.3 [0; 7.2]	0.565
Spinal cord ischemia, n (%)	2 11% [3%; 31%]	0	RD: 11% [−3%; 24%] OR: 0 [0; 3.8]	0.173
Lung ventilation, hours (range)	23 [17; 50] (6–1654)	24.5 [16.75; 50.75] (4–831)	pseudo-MED (95%CI): 0.95 [−11; 10] SMD (95%CI): 0.5 [−0.14; 1.14]	0.966
Respiratory failure (tracheostomy), n (%)	4 21% [9%; 43%]	6 23% [11%; 42%]	RD: −2% [−26%; 22%] OR: 1.1 [0.2; 6.4]	>0.999
Myocardial infarction, n (%)	2 11% [3%; 31%]	2 8% [2%; 24%]	RD: 3% [−14%; 20%] OR: 0.7 [0; 10.8]	>0.999
Creatinine (24 h after surgery), mmol/L	134 [101.25; 154.75]	180.5 [116; 257]	pseudo-MED (95%CI): 40 [0; 106] SMD (95%CI): −0.77 [−1.44; −0.1]	0.054
AKI (dialysis), n (%)	3 16% [6%; 38%]	8 31% [17%; 50%]	RD: −15% [−39%; 9%] OR: 2.3 [0.5; 16]	0.309
MOF, n (%)	2 11% [3%; 31%]	6 23% [11%; 42%]	RD: −13% [−34%; 9%] OR: 2.5 [0.4; 28.5]	0.435
Blood loss, mL	500 [300; 600]	600 [500; 700]	pseudo-MED (95%CI): 100 [0; 300] SMD (95%CI): −0.57 [−1.22; 0.08]	0.072
Bleeding (re-sternotomy), n (%)	0	4 15% [6%; 34%]	-	0.126
RBC transfusion, unit	2 [1; 2]	2 [2; 3]	-	0.715
FFP transfusion, unit	3 [2; 3]	3 [2; 4]	-	0.441
Platelet transfusion, unit	2 [1; 2]	2 [2; 3]	-	0.888
Deep/superficial wound complications, n (%)	0	0	-	>0.999
30-day mortality, n (%)	0	2 8% [2%; 24%]	-	0.501

AKI—acute kidney injury, FFP—fresh frozen plasma, ICU—intensive care unit, MODS—multiple organ failure, RBC—red blood cell, pseudo-MED—pseudo-median-based procedure for comparing two independent groups, RD—risk difference, SMD—standardized mean difference.

## Data Availability

The datasets used and/or analyzed during the current study are available from the corresponding author upon reasonable request.

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
