# Peer review of "Frozen Elephant Trunk for Aortic Dissection Using Different Hybrid Grafts: Preliminary Results from a Prospective Study"

_jpm, 2023, doi:10.3390/jpm13050784_

Round 1
Reviewer 1 Report
In your paper the results after treatment with FET in acute(20)and chronic(25) dissections are reported together: this is not correct and the 2 types should be distinguished.
In the materials and method only 1 case of SCI is reported while in the discussion you speak of 2 case after E-vita graft.
A higher incidence of renal failure is reported (31vs16%)in the Med Eng group;why since this graft has the possibility of distal perfusion of the aorta compared to E-Vita graft?
The discussion is a bit lacking; the characteristic of the 2 graft used should be better described in relation to the prevention of the development of complications such as SCI,renal failure as well as neurological complications.
In the conclusions ,given the limitations of the study,it should be reported that the short term results seem superimposable.Regarding the higher mortality in the medium term in the E-Vita Group,it is not possible to draw this conclusion because the reasons for the deaths are independent of the type of the graft implanted.
Author Response
Dear Reviewer, we would like to thank you for all the comments and suggestions.
All our responses we uploaded in a separate file.

Reviewer 2 Report
The article by Boris Kozlov et al “Frozen elephant trunk for aortic dissection using different hybrid grafts: preliminary results from a prospective study” is well-designed and the results appears to be well-analyzed.
Major Comments:
P1, Line 31 – The author should provide information on incidence and prognosis of aortic dissection, as well as the current treatment options.
P1, Line 39 – The author has mentioned that various modifications of hybrid grafts for the FET are used – but not provided fully explanation and also should mention limitations of existing grafts.
P2, Line 47 – The author should clearly define the inclusion and exclusion criteria including the specific types of aortic dissection included in the study.
P2, Line 50 – The authors should provide how the patients were randomized to receive different graft types.
P2, Line 57 – The authors should provide more information on the follow-up period and how outcomes were assessed.
P3, Line 122 – The authors should discuss whether there were any subgroups of patients who benefited more or less from each graft type.
P10, Line 192 – The author could discuss how the study results may inform decisions about which graft type to use in different patients, or what future research directions may be needed.
Minor Comments:
The author should proofread and improve the clarity and flow of the writing.
The author need to do proofreading and editing since the flow is confusing.
Author Response

(The authors gave the same response as above.)
